# Production, Characterization Purification, and Antitumor Activity of L-Asparaginase from *Aspergillus niger*

Suzane Meriely da Silva Duarte [1], Allysson Kayron de Carvalho Silva [1], Katia Regina Assunção Borges [1], Carolina Borges Cordeiro [2], Fernanda Jeniffer Lindoso Lima [1], Marcos Antônio Custódio Neto da Silva [3,*], Marcelo de Souza Andrade [1,2] and Maria do Desterro Soares Brandão Nascimento [1,2]

[1] Programa de Doutorado em Biotecnologia—Rede Nordeste de Biotecnologia (RENORBIO), Center for Basic and Applied Immunology (NIBA), Federal University of Maranhão, São Luís 65080-805, Maranhão, Brazil; suzanemeriely25@gmail.com (S.M.d.S.D.); allysson.carvalho@discente.ufma.br (A.K.d.C.S.); borges.katia@discente.ufma.br (K.R.A.B.); fernanda.jeniffer@discente.ufma.br (F.J.L.L.); marcelo.andrade@ufma.br (M.d.S.A.)
[2] Programa de Pós-Graduação em Saúde do Adulto (PPGSAD), Center for Basic and Applied Immunology (NIBA), Federal University of Maranhão, São Luís 65080-805, Maranhão, Brazil; carolinacbc23@gmail.com
[3] Departament of Medicina I, Federal University of Maranhão, Imperatriz 65915-060, Maranhão, Brazil
[*] Correspondence: marcos.antonio@ufma.br

**Abstract:** Cervical cancer is caused by a persistent and high-grade infection. It is caused by the Human Papillomavirus (HPV), which, when entering cervical cells, alters their physiology and generates serious lesions. HPV 18 is among those most involved in carcinogenesis in this region, but there are still no drug treatments that cause cure or total remission of lesions caused by HPV. It is known that L-asparaginase is an amidohydrolase, which plays a significant role in the pharmaceutical industry, particularly in the treatment of specific cancers. Due to its antitumor properties, some studies have demonstrated its cytotoxic effect against cervical cancer cells. However, the commercial version of this enzyme has side effects, such as hypersensitivity, allergic reactions, and silent inactivation due to the formation of antibodies. To mitigate these adverse effects, several alternatives have been explored, including the use of L-asparaginase from other microbiological sources, which is the case with the use of the fungus *Aspergillus niger*, a high producer of L-asparaginase. The study investigated the influence of the type of fermentation, precipitant, purification, characterization, and in vitro cytotoxicity of L-asparaginase. The results revealed that semisolid fermentation produced higher enzymatic activity and protein concentration of *A. niger*. The characterized enzyme showed excellent stability at pH 9.0, temperature of 50 °C, resistance to surfactants and metallic ions, and an increase in enzymatic activity with the organic solvent ethanol. Furthermore, it exhibited low cytotoxicity in GM and RAW cells and significant cytotoxicity in HeLa cells. These findings indicate that L-asparaginase derived from *A. niger* may be a promising alternative for pharmaceutical production. Its attributes, including stability, activity, and low toxicity in healthy cells, suggest that this modified enzyme could overcome challenges associated with antitumor therapy.

**Keywords:** cervical cancer; L-asparaginase; *Aspergillus niger*

## 1. Introduction

Cervical cancer is caused by a persistent and high-grade infection, which can be caused by the Human Papillomavirus (HPV), which, upon entering cervical cells, alters their physiology and generates serious lesions [1]. It is worth mentioning that, to date, there are no drug treatments that cause cure or total remission of lesions caused by HPV [2]. HPV 18 is among those most involved in carcinogenesis in this region, which is why the use of HeLa cells in this study is justified since they are cervical carcinoma cells transformed by HPV 18 [3].

It is known that L-asparaginase is an amidohydrolase [4], which plays a significant role in the pharmaceutical industry, particularly in the treatment of specific cancers, due to its antitumor properties. Some studies have demonstrated its cytotoxic effect against HeLa cells [5]. This enzyme is industrially produced and commercially available, constituting 40% of global enzyme demand, representing USD 380 million in sales in 2017. This number is predicted to increase to USD 420 million by 2025 [6]. L-asparaginase, originating from bacteria, is available in three formulations: two from *Escherichia coli* and one from *Erwinia chrysanthemi* [7].

Despite its therapeutic applications, commercially available L-asparaginase has side effects such as hypersensitivity, allergic reactions, silent inactivation due to the formation of anti-asparaginase antibodies, and a short serum half-life [8]. These problems arise in part from their microbiological origin and strong immunological response [9]. To minimize adverse effects, several alternatives have been explored, including L-asparaginase from different sources, protein engineering, and enzyme immobilization [10].

*Aspergillus niger*, a vital industrial fermentation strain, is crucial in biotechnology. Its metabolic compounds are widely used in the fermentation of animal feed, food additives, industrial enzyme preparations, and biotransformation [11–15]. Due to its ability to produce enzymes in high concentrations and beneficial pharmaceutical supplies, *A. niger* is a focus of biotechnological research [16,17].

Given these considerations, the search for a new, highly efficient, and stable source of L-asparaginase for pharmaceutical and biotechnological applications continues. This study employed strategies to optimize the production of L-asparaginase through fermentation using *A. niger*, aiming to explore its catalytic properties to achieve a stable enzyme with high activity and efficiency and test its antitumor activity in HeLa cells.

## 2. Materials and Methods

### 2.1. Materials

EDTA (TBE) was purchased from Ludwig Biotechnology (São Paulo, SP, Brazil); Methanol and Acetone were purchased from Vetec (Rio de Janeiro, RJ, Brazil); Isopropanol and Sodium Hydroxide were acquired from Quimex (São Paulo, SP, Brazil); Aluminum Sulfate ($AlSO_4$), Barium Chloride ($BaCl_2$), Copper Sulfate ($CuSO_4$), Calcium Chloride ($CaCl_2$), Magnesium Chloride ($MgCl_2$), Iron Sulfate ($FeSO_4$), Potassium Chloride ($KCl$), Zinc Sulfate ($ZnSO_4$), Iron Chloride ($FeCl$), Manganese Chloride ($MnCl_2$), Peptone, Citric Acid, Potassium Phosphate, and Tween 20 and 80 were procured from ISOFAR (São Paulo, SP, Brazil); Dipotassium Phosphate, L-asparagine, Hydrochloric Acid (HCl), Triton-X100, Tris(hydroxymethyl)aminomethane (TRIS), Sodium Dodecyl Sulfate (SDS), Ammonium Sulfate (($NH_4)_2SO_4$), Ethylenediaminetetraacetic Acid (EDTA), Sodium Chloride (NaCl), Human Fibroblast (GM), and Macrophage (RAW 264.7) cell lines were purchased from Sigma-Aldrich (São Paulo, SP, Brazil). Human cancer cells, HeLa, were used (cervix adenocarcinoma; ATCC CCL-2).

### 2.2. Comparative Study of Different Fungi in the Production of L-Asparaginase

With the aim of identifying potential sources of production and high enzymatic activity of L-asparaginase, we examined the enzymatic activities of crude extracts produced by *A. niger*, *Aspergillus flavus*, *Curvulária*, *Fusarium solani*, *Fusarium oxysporum*, *Penicillium decumbens*, and *Rhizopus* sp. All microorganisms in the study were obtained from the mycotheque at the Federal University of Maranhão, São Luís, Brazil. The production of L-asparaginanse in these microorganisms was investigated through two types of fermentation: solid (SsF) and submerged (SmF).

#### 2.2.1. Solid-State Fermentation (SsF) of L-Asparaginase

For the production of L-asparaginase using SsF technique, the methodology described by Dias et al. [18] was employed with some adaptations. For each microorganism, a mixture of 50 g of wheat bran with 20 mL of 100 mMol/$L^{-1}$ L-asparagine solution diluted in

50 mM Tris-HCl buffer at pH 8.0 was prepared. The mixture was sterilized, and approximately 5 fragments of the microorganism culture medium, each about 5 mm$^2$, were added. Fermentation was maintained at a temperature of 30 °C for 120 h.

After the fermentation period, the cultures were treated with 50 mL of 50 mMol/L$^{-1}$ Tris-HCl buffer at pH 8.0. Agitation was carried out at 190 rpm at a temperature of 25 °C for 60 min. Subsequently, the cultures were filtered and centrifuged at 14,000× $g$ rpm at 4 °C, and the supernatant was considered as the crude extract of SsF.

### 2.2.2. Submerged Fermentation (SmF) of L-Asparaginase

For SmF, the methodology described by Mahajan et al. [19] was used with adaptations. Flasks containing 50 mMol/L$^{-1}$ Tris-HCl buffer at pH 8.0 were supplemented with NH$_4$NO$_3$ (2 g/L), KH$_2$PO$_4$ (1.52 g/L), KCl (0.52 g/L), MgSO$_4$ (0.52 g/L), CuSO$_4$ (0.001 g/L), ZnSO$_4$ (0.001 g/L), FeSO$_4$ (0.001 g/L), and L-asparagine (10 g/L). Approximately 5 fragments of the culture medium, each about 5 mm$^2$, were added. The liquid media were kept under orbital agitation at 190 rpm and a temperature of 30 °C for 7 days. Samples were collected every 24 h to determine the time of maximum enzymatic activity. The collected samples were filtered and centrifuged at 10,000× $g$ rpm at 4 °C for 25 min, and the supernatant was considered as the crude extract of SmF.

### 2.3. Fungal Culture and Isolation

The environmental sample of the filamentous fungus used for these analyses, previously identified as belonging to the species *A. niger*, was obtained from the mycoteca at the Federal University of Maranhão, São Luís, Maranhão, Brazil. Fungal cultivation was carried out in Petri dishes of 4% Sabouraud Dextrose culture medium (MERCK) and incubated in a Bacteriological Oven (SOLAB SL-101, São Paulo, SP, Brazil) at 37 °C for 5 days. All cultivation and subculture procedures took place in a laminar flow hood close to the Bunsen burner.

### 2.3.1. DNA Extraction

The biochemical DNA extraction protocol was used following the methodology of Valenzuela-Lopez et al. [20] with the addition of glass spherules and adaptations to perform cell lysis of the chitin wall. Subsequently, the quality of the DNA was checked, followed by DNA quantification and purity using a Nanodrop One C, Waltham, MA, USA). The samples contained between 100 and 200 ηg/μL and were sent for amplification through polymerase chain reaction.

### 2.3.2. Primers and Polymerase Chain Reaction (PCR)

For molecular analysis, species-specific primers (Thermofisher-Scientific, Waltham, MA, USA) of the internal transcribed spacer region (ITS) and ITS1-5.8S-ITS2 fragments were used [21]. The nucleotide sequences used were ITS 1 (5′-GCTCATTAAATCAGTTATCG-3′) and ITS 2 (5′-GTTATTATGATTCACCAAGG-3′-3′) according to Alabdalall et al. [22].

Polymerase chain reaction (PCR) of the regions had a final volume of 25 uL, using a MasterMix PCR Set (Ludwig Biotechnology, Baltimore, MD, USA) containing Tris-KCl, pH 8.4; 2.0 mM of MgCl$_2$; 0.2 mM of DNTP mix, and 2.5 U of Taq DNA polymerase; 12.5 μL of Pre-Mix; 6.5 μL of ultra-pure water; 2.5 μL of ITS1-5.8S-ITS2 primers (Termofisher Invitrogen, Carlsbad, CA, USA, 10 pmol mL$^{-1}$), 1.0 μL of DNA (5 ng mL$^{-1}$), and following the following thermocycling conditions in the Biocycler MJ96G Thermocycling equipment (Hercules, CA, USA)

The PCR cycles performed for *A. niger* samples were 95 °C for 8 min, 34 cycles at 94 °C for 1 min, 57 °C for 1 min, and 72 °C for 1 min and then 72 °C for 7 min for extension and finishing at 4 °C. The amplicons were subjected to electrophoresis in 1.5% agarose gel at 90 V for 120 min in TBE 1X (Ludwig Biotechnology, Baltimore, MD, USA) and molecular marker (Ladder 100 bp, 0.1 μg/μL, Ludwig Biotechnology, Baltimore, MD, USA). The

products were evaluated for quality using a Transluminator (Loccus L-PIX TOUCH, Vilber Lourmat ECX-F20.M).

### 2.4. Purification of Crude L-Asparaginase Extracts

The crude extracts from SmF and SsF were precipitated according to the methodology of Vala et al. [23], adapted for different fractions of L-asparaginase: 0–40% and 40–80% ethanol, isopropanol, and ammonium sulfate, aiming to minimize significant protein losses and determine the most suitable precipitation method to optimize L-asparaginase extraction. The precipitants were added to the crude L-asparaginase extracts, which were kept at 4 °C for 24 h with gentle agitation and then centrifuged at $14,000 \times g$ rpm for 25 min at 4 °C.

The precipitates were resuspended in a volume corresponding to 1/4 of the total volume of the centrifuged extract, using 50 mMol/L$^{-1}$ of Tris-HCl buffer at pH 7.0. Subsequently, dialysis was performed; the suspension was dialyzed against the same buffer used for crude extract preparation at 4 °C for 6 h, with two buffer changes every 3 h, with the last change kept overnight. After dialysis, the precipitates were centrifuged at $10,000 \times g$ rpm for 25 min at 4 °C and resuspended in the same buffer used for extract preparation. They were then stored at −34 °C for subsequent studies. These fractions were classified as L-asparaginase F1 (fraction 0–40%) and L-asparaginase F2 (fraction 40–80%). Fraction purifications were performed using ion exchange chromatography utilizing a diethylaminoethyl cellulose (DEAE cellulose) column.

### 2.5. Protein Concentration

Protein concentration was estimated using the method described by Warburg and Christian [24] with a spectrophotometer at absorbances of 260 nm and 280 nm, using the following formula:

$$[Protein\ Concentration] = A260\,\text{nm} \times 1.55 - A280\,\text{nm} \times 0.75 \tag{1}$$

### 2.6. Determination of L-Asparaginase Activity

L-asparaginase activity was assessed by the formation of β-hydroxamate aspartic acid from asparagine and hydroxylamine, following the method described by Drainas et al. [25]. The reaction mixture consisted of 300 μL of 20 mmol L$^{-1}$ of Tris-HCl buffer at pH 8.0, 100 μL of 100 mmol L$^{-1}$ of L-asparagine, 100 μL of 1 M of hydroxylamine, and 500 μL of the enzyme sample. The mixture was incubated in a water bath at 37 °C for 15 min, and the reaction was stopped by adding 250 μL of a solution containing HCl (2.4%), ferric chloride (10%), and TCA (5%).

The enzymatic reaction between β-hydroxamate aspartic acid and ferric chloride produced a reddish color, and its absorbance was measured at 500 nm using a spectrophotometer. An analytical curve was constructed using a solution of β-hydroxamate aspartic acid (0.1 μmol/mL to 3 μmol/mL). One unit of L-asparaginase activity (U) was defined as the amount of enzyme required to form one μmol of β-hydroxamate aspartic acid per minute under the assay conditions. The specific activity of L-asparaginase was expressed as μmol of β-hydroxamate aspartic acid formed per minute per milligram of protein.

### 2.7. Biochemical Characterization of Purified L-Asparaginase from Fungus Extract
#### 2.7.1. Fungi Optimal pH and Stability

Different pH ranges from 3.0 to 9.0 were tested using 0.1 M of acetate buffer, 0.1 M of Tris-HCl, and 0.1 M of citrate–phosphate solutions. For stability, the reaction at each pH was maintained for 1 h at 37 °C, and the tests were performed in triplicate.

2.7.2. Optimal Temperature and Stability

To determine the optimal temperature, L-asparaginase solution was incubated for 30 min at temperatures ranging from 10 to 90 °C. For stability, the enzyme was kept for 1 h at the same temperature ranges. The tests were performed in triplicate.

2.7.3. Effect of Surfactants and Metal Ions in Salts

To evaluate stability against different surfactants and ions, the method adapted from Vala et al. [23] was used. L-asparaginase was incubated for 1 h with different proportions of non-ionic surfactants: Triton X-100, Tween-20, and Tween-80 (0.01%, 0.10%, and 0.50% *w/v*) and ionic surfactant sodium dodecyl sulfate (SDS). The effect of metal ions on activity was determined by pre-incubating L-asparaginase in 50 mM of Tris-HCl buffer (pH 8.0) with 0.01 mM of $CoSO_4$, $FeCl_3$, $CaCO_3$, $NaCl$, $MgCl_2$, $MnSO_4$, and $ZnSO_4$ at 35 °C for 1 h. The tests were performed in triplicate, and residual activity was measured.

2.7.4. Stability in Organic Solvents

To determine the stability of *A. niger* L-asparaginase in acetone, ethanol, methanol, and isopropanol, the method adapted from Oliveira et al. [26] was used. The enzyme was incubated for 1 h at 37 °C in different solvent concentrations (25%, 50%, 80%, and 100% *v/v*). After incubation, residual activity was measured.

*2.8. Cytotoxic Activity of L-Asparaginase*

The cytotoxicity of the enzyme was evaluated using the methods described by El-Gendy et al. [27] with adaptations. L-asparaginase was tested at concentrations ranging from 100 to 6.25 μg/mL$^{-1}$ against cells derived from human fibroblasts (GM cells), macrophage cell line (RAW 264.7), and HeLa (human cervical cancer). Cell viability was determined using 3-(4,5-dimethylthiazol-2-yl)-2,5-diphenyltetrazolium bromide (MTT) as follows:

$$[(Sample - Blank)/(Control - Blank) \times 100] \tag{2}$$

*2.9. Morphological Viability in HeLa Cells by Inverted Light Microscopy*

Cellular morphology was analyzed by inverted light microscopy Optika Microscope (Ponteranica, Lombardia, ITA) after treatment with L-asparaginase at a concentration of 100 μg/mL. Image analysis was performed using Axiovision Release 4.8.1 software (Carl Zeiss Inc., Jena, Germany). Cells were cultured in 12-well plates in the presence and absence of L-asparaginase for 24, 48, and 72 h and then observed under a microscope.

*2.10. Statistical Analysis*

The results of this study were analyzed using ANOVA and Tukey's post hoc test ($p < 0.05$) with GraphPad Prism 8.0.1 software. The results were reported as mean ± standard deviation (SD) of triplicate determinations.

**3. Results**

*3.1. Tracking and Optimization of L-Asparaginase Production by Fungi*

3.1.1. Screening for L-Asparaginase Producers

The results indicated that, through the SsF process, the studied fungi exhibited higher L-asparaginase production, suggesting that this method was the most suitable for enzyme production. Among the tested species, *A. niger* demonstrated the highest activity, 46.4 U/mL, compared to other fungal species: *Aspergillus flavus* (30.23 U/mL), *Curvularia* (23.41 U/mL), *Fusarium solani* (37.64 U/mL), *Fusarium oxysporum* (21.69 U/mL), *Rhizopus* sp. (11.75 U/mL), and *Penicillium decumbens* (24.09 U/mL) (Figure 1). Due to its higher enzymatic activity, *A. niger* L-asparaginase was further studied in subsequent steps.

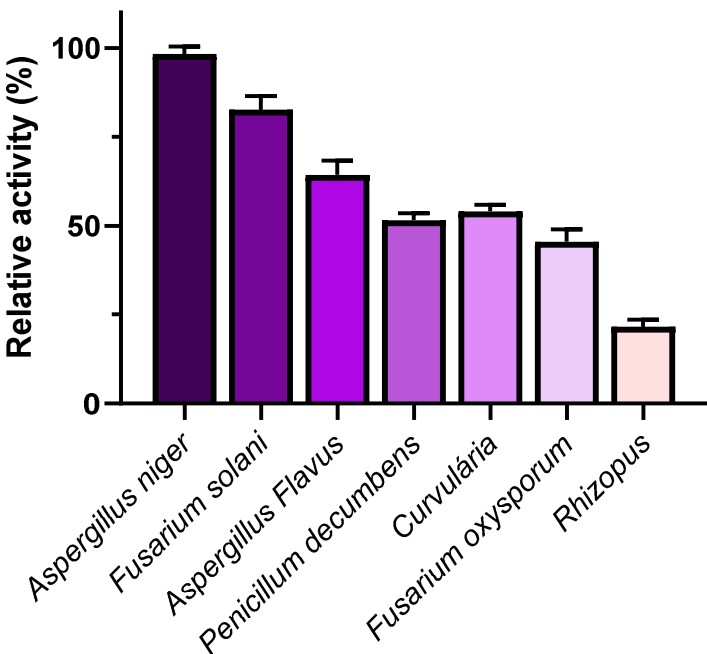

**Figure 1.** L-asparaginase activity produced by different fungi. Results are presented as relative activity ± standard deviation; each measurement was performed in triplicate using asparagine as a substrate at 37 °C for 15 min.

### 3.1.2. Fermentation Type for L-Asparaginase Production

In order to achieve high enzymatic activity and concentration of L-asparaginase produced by *A. niger*, the production of this enzyme was analyzed in two types of fermentation. SsF exhibited higher activity in all stages of purification. Consequently, L-asparaginase showed superior activity compared to SmF in the 0–40% fraction and 40–80% fraction stages, displaying specific activity (187.19 U/mg) with yields of 52.25% and purification of 17.08 in the 40–80% fraction, respectively, compared to the SmF fractions (Table 1).

**Table 1.** Partial purification results of L-asparaginase from *A. niger* from submerged fermentation (SmF) and solid-state fermentation (SsF). Total protein and enzymatic activities were determined according to Warburg and Christian [24] and Drainas et al. [25], respectively, using L-asparagine as substrate. Samples included crude extract, Fraction 1 (F1 (0–40%)), and Fraction 2 (F2 (0–80%)) from submerged fermentation (SmF) and solid-state fermentation (SsF).

| Type of Fermentation | Purification Stage | Total Activity (U/mL) | Total Protein (mg) | Specific Activity (U/mg) | Yield (%) | Purification |
|---|---|---|---|---|---|---|
| SmF | Crude Extract | 3711.50 | 480.00 | 7.73 | 100 | 1 |
| | Fraction 1 (0–40%) | 363.96 | 18.95 | 19.21 | 9.81 | 2.48 |
| | Fraction 2 (40–80%) | 767.20 | 44.93 | 17.07 | 20.67 | 2.21 |
| SsF | Crude Extract | 5261 | 480 | 10.96 | 100.00 | 1.00 |
| | Fraction 1 (0–40%) | 888.58 | 84.63 | 10.50 | 16.89 | 0.96 |
| | Fraction 2 (40–80%) | 2748.9 | 14.685 | 187.19 | 52.25 | 17.08 |

### 3.2. Molecular Identification of A. niger

After molecular analysis, amplification of the sample resulted in a band of approximately 295 bp (base pairs), which corroborates what is described in the literature for the species (Figure 2).

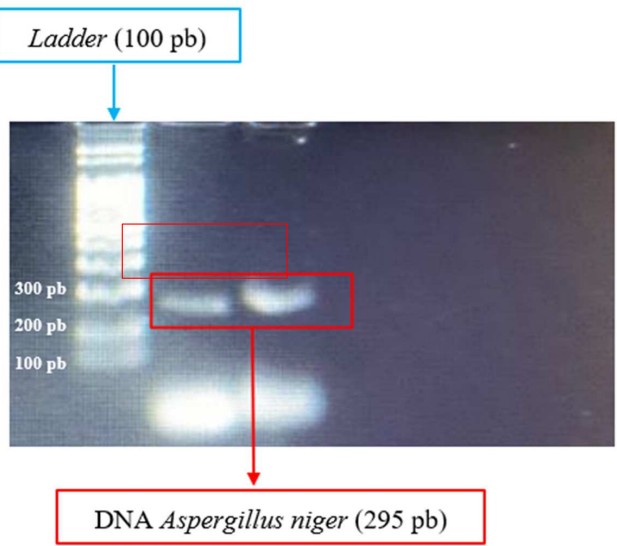

**Figure 2.** The 1.5% agarose gel with *A. niger* samples (fragments highlighted in red), demonstrating approximately 295 bp.

### 3.3. Precipitant Type for L-Asparaginase Purification

Since solid-state fermentation (SsF) exhibited higher specific activity of L-asparaginase and a higher protein concentration per milliliter of extract produced by *A. niger*, purification and optimization of the precipitation method were performed using different types of precipitants to determine which one enhances the concentration of L-asparaginase. The results showed that ethanol, when used as a protein precipitant to obtain L-asparaginase produced by *A. niger*, led to a higher concentration of L-asparaginase in the 40–80% fraction, with a higher specific activity of 34.35 U/mg, compared to isopropanol (12.33 U/mg) and ammonium sulfate (6.39 U/mg) used as precipitants (Table 2).

**Table 2.** Influence of precipitation agents in *A. niger* solid-state fermentation (SsF). Total protein and enzymatic activities were determined, respectively, according to Warburg and Christian [24] and Drainas et al. [25], using L-asparagine as a substrate. Samples included crude extract, Fraction 1 (0–40%), Fraction 2 (0–80%), and DEAE cellulose column fractions from solid-state fermentation (SsF).

| Type of Precipitation | Purification Stage | Total Activity (U/mL) | Total Protein (mg) | Specific Activity (U/mg) | Yield | Purification (%) |
|---|---|---|---|---|---|---|
| Ethanol | Crude Extract | 5520.50 | 3367.5 | 1.64 | 100.00 | 1.00 |
| | Fraction 1 (0–40%) | 1205.68 | 84.63 | 14.25 | 21.84 | 8.69 |
| | Fraction 2 (40–80%) | 2671.90 | 114.18 | 23.40 | 48.40 | 14.27 |
| | DEAE Cellulose Column | 1092.30 | 31.8 | 34.35 | 19.79 | 20.95 |
| Ammonium Sulfate | Fraction 1 (0–40%) | 182.16 | 57.09 | 3.19 | 3.30 | 1.95 |
| | Fraction 2 (40–80%) | 826.20 | 152.01 | 5.44 | 14.97 | 3.32 |
| | DEAE Cellulose Column | 265.00 | 21.5 | 12.33 | 4.80 | 7.52 |
| Isopropanol | Fraction 1 (0–40%) | 113.40 | 64.35 | 1.76 | 2.05 | 1.07 |
| | Fraction 2 (40–80%) | 306.60 | 143.955 | 2.13 | 5.55 | 1.30 |
| | DEAE Cellulose Column | 104.10 | 16.3 | 6.39 | 1.89 | 3.90 |

### 3.4. Biochemical Characterization of L-Asparaginase from A. niger

The biochemical characterization of L-asparaginase precipitated by ethanol was conducted to assess the stability of the enzyme's catalytic activity under various conditions, aiming to apply the enzyme in biochemical processes effectively.

#### 3.4.1. Optimal pH and Stability

The enzymatic activity of L-asparaginase produced by *A. niger* was tested across the pH range of 3.0 to 9.0. The results indicated that the optimal pH for L-asparaginase

activity was pH 9.0, with an enzymatic activity of 47.25 U/mL (Figure 3A). The enzyme exhibited activity above 80% in the pH range of 5.5 to 8.5 compared to its optimal pH. Additionally, the enzyme did not exhibit significant resistance to different pH ranges in this study, differing from the findings in previous research (Figure 3A). In terms of pH stability (Figure 3B), the L-asparaginase produced by *A. niger* demonstrated maximum activity at pH 9.0, consistent with its optimal pH.

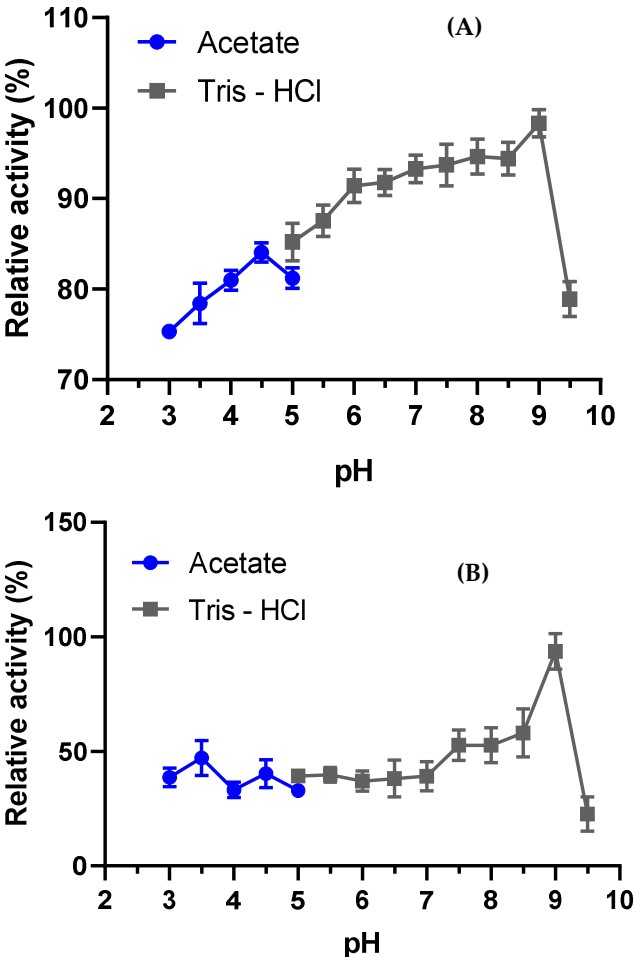

**Figure 3.** Optimal pH (**A**) and stability (**B**) of *A. niger* L-asparaginase. Results are presented as relative activity ± standard deviation. Enzymatic activity was measured using asparagine as a substrate at 37 °C for 15 min to determine the optimal pH and for 1 h to assess stability. All tests were conducted in triplicate.

### 3.4.2. Optimal Temperature and Stability

The L-asparaginase produced by *A. niger* exhibited maximum activity at 50 °C and demonstrated resistance to other temperatures within the range of 30 to 45 °C (Figure 4A). Regarding temperature stability, the enzyme maintained its maximum activity up to 50 °C, similar to its optimal temperature. Subsequently, there was a decline in activity, leading to complete loss of enzymatic activity, as depicted in Figure 4B. These results highlight the enzyme's preference for higher temperatures but also indicate a limited stability range beyond its optimal temperature.

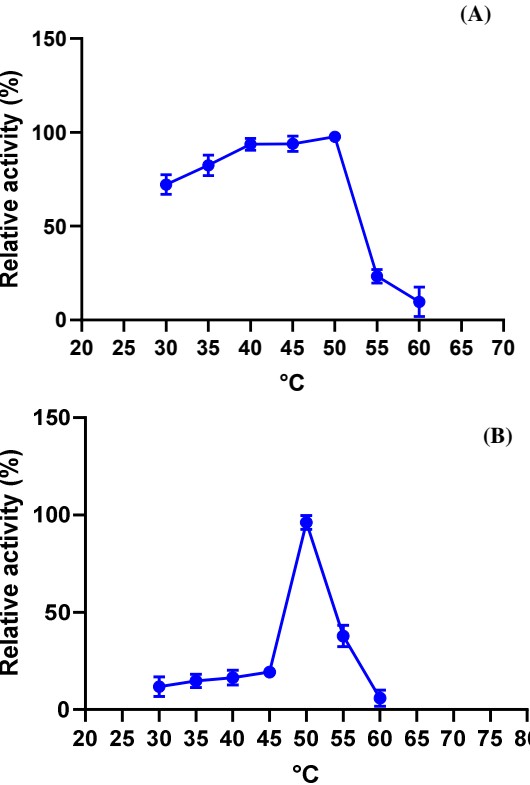

**Figure 4.** Optimal temperature (**A**) and stability (**B**) of *A. niger* L-asparaginase. Results are presented as relative activity ± standard deviation. Enzymatic activity was measured using asparagine as a substrate for 15 min to determine the optimal temperature and for 1 h to assess stability. All tests were conducted in triplicate.

### 3.4.3. Effect of Surfactants and Metal Ions in Salts

The objective of this study was to understand the L-asparaginase produced by *A. niger* and identify compounds that might cause a decrease in enzymatic activity. Various surfactants and metal ions were tested to assess their impact on the enzyme's activity, which is crucial for its application in pharmaceutical compounds and biotechnological processes. The residual activity of *A. niger*-produced L-asparaginase was examined concerning different surfactants. There was a significant increase in enzymatic activity with Tween 20, showing a 16.41% increase at a concentration of 0.10% and an 8.15% increase at a concentration of 0.50%. However, in the case of EDTA, enzymatic activity decreased at a higher concentration of 0.50%, with a residual activity of 40.29% (Table 3). These findings provide valuable insights into the enzyme's response to various surfactants and metal ions, aiding in its tailored application in specific contexts.

**Table 3.** Effect of surfactants on the activity of *A. niger* L-asparaginase. The percentage values correspond to the amount of surfactant in 50 mMol of phosphate buffer at pH 7.0. Tests were conducted in triplicate. L-asparaginase samples were mixed with surfactants (0.01%, 0.10%, and 0.50% *w/v*) and incubated for 1 h at 37 °C before determining residual activity. Results are presented as enzymatic activity ± standard deviation.

| Type | Concentrations | U/mL | Relative Activity (%) ± SD |
|---|---|---|---|
| Control | | 47.48 | 100 ± 0.014 |
| Tween 20 | 0.01% | 44.98 | 94.73 ± 0.035 |
| | 0.10% | 55.27 | 116.41 ± 0.084 |
| | 0.50% | 51.35 | 108.15 ± 0.028 |

**Table 3.** *Cont.*

| Type | Concentrations | U/mL | Relative Activity (%) $\pm$ SD |
|---|---|---|---|
| Triton X-100 | 0.01% | 45.91 | 96.69 $\pm$ 0.013 |
| | 0.10% | 36.56 | 77.00 $\pm$ 0.018 |
| | 0.50% | 33.92 | 71.44 $\pm$ 0.020 |
| Tween 80 | 0.01% | 41.61 | 87.64 $\pm$ 0.011 |
| | 0.10% | 42.45 | 89.41 $\pm$ 0.006 |
| | 0.50% | 38.01 | 80.05 $\pm$ 0.027 |
| SDS | 0.01% | 35.02 | 73.76 $\pm$ 0.012 |
| | 0.10% | 24.13 | 50.82 $\pm$ 0.009 |
| | 0.50% | 24.43 | 51.45 $\pm$ 0.025 |
| E.D.T. A | 0.01% | 31.1 | 65.50 $\pm$ 0.005 |
| | 0.10% | 22.12 | 46.59 $\pm$ 0.004 |
| | 0.50% | 19.13 | 40.29 $\pm$ 0.003 |

In the presence of ions, the enzymatic activity of L-asparaginase exhibited a 53.94% increase when in contact with $Co^{2+}$ and a 48.20% increase with $Mg^{2+}$. Conversely, the enzyme's activity decreased by 52.67% upon contact with $Ca^{2+}$ (Table 4).

**Table 4.** Influence of ions on the activity of *A. niger* L-asparaginase. The L-asparaginases were incubated with ions (1 mM) in 50 mM of Tris-HCl buffer at pH 7.0 and then incubated for 1 h at 37 °C before determining residual activity. Tests were conducted in triplicate. Results are presented as residual activity $\pm$ standard deviation.

| Type | U/mL | Relative Activity (%) $\pm$ SD |
|---|---|---|
| Control | 31.31 | 100 $\pm$ 0.04 |
| $Mg^{2+}$ | 46.4 | 148.20 $\pm$ 0.02 |
| $Co^{2+}$ | 16.89 | 153.94 $\pm$ 0.01 |
| $SO_4^{2-}$ | 15.05 | 48.07 $\pm$ 0.03 |
| $Zn^{+2}$ | 16.21 | 51.77 $\pm$ 0.01 |
| $Ca^{2+}$ | 46.13 | 47.33 $\pm$ 0.01 |
| $Fe^{3+}$ | 27.63 | 88.25 $\pm$ 0.01 |
| $Na^+$ | 28.47 | 90.93 $\pm$ 0.02 |

### 3.4.4. Stability in Organic Solvents

L-asparaginase exhibited increased enzymatic activity when in contact with organic solvents such as ethanol, showing a 46.24% increase in activity at a concentration of 50% (*v/v*). Similar trends were observed with isopropanol, where a 41.41% increase was observed at a concentration of 25% (*v/v*), and with methanol, which displayed the highest activity, showing a 77.20% increase at a concentration of 100% (Table 5).

**Table 5.** Effect of organic solvents on *A. niger* L-asparaginase. The percentage values associated with polar organic solvents correspond to the amount of solvent in 50 mMol/L of phosphate buffer, pH 7.0. Tests were conducted in triplicate. L-asparaginase samples were mixed with organic solvents (25%, 50%, 80%, and 100% *v/v*) and incubated for 1 h at 37 °C before determining residual activity. Results are presented as enzymatic activity $\pm$ standard deviation.

| Type | Concentrations | U/mL | Relative Activity (%) $\pm$ SD |
|---|---|---|---|
| Control | | 32.48 | 100 $\pm$ 0.014 |
| Acetone | 25 | 16.79 | 51.72 $\pm$ 0.05 |
| | 50 | 14.53 | 44.72 $\pm$ 0.02 |
| | 80 | 5.76 | 17.73 $\pm$ 0.03 |
| | 100 | 2.49 | 7.67 $\pm$ 0.04 |
| Ethanol | 25 | 24.15 | 74.38 $\pm$ 0.04 |
| | 50 | 15.02 | 46.18 $\pm$ 0.01 |
| | 80 | 9.54 | 29.34 $\pm$ 0.02 |
| | 100 | 6.28 | 19.33 $\pm$ 0.02 |

**Table 5.** *Cont.*

| Type | Concentrations | U/mL | Relative Activity (%) $\pm$ SD |
|---|---|---|---|
| Isopropanol | 25 | 13.45 | 41.37 $\pm$ 0.03 |
| | 50 | 10.26 | 31.59 $\pm$ 0.01 |
| | 80 | 6.75 | 20.78 $\pm$ 0.03 |
| | 100 | 4.14 | 12.75 $\pm$ 0.01 |
| Methanol | 25 | 8.55 | 26.32 $\pm$ 0.09 |
| | 50 | 7.82 | 24.05 $\pm$ 0.03 |
| | 80 | 5.19 | 16.00 $\pm$ 0.02 |
| | 100 | 2.34 | 7.20 $\pm$ 0.07 |

*3.5. Cytotoxic Activity of L-Asparaginase in GM, RAW, and HeLa Cells*

The cytotoxic activity of L-asparaginase from *A. niger* was evaluated in GM, RAW, and HeLa cells (Figure 5). These cells were treated with 6.25 up to 100 µg/mL of the enzyme at 24, 48, and 72 h. L-asparaginase in GM and RAW cells showed a slight cytotoxic activity at 24 and 48 h only at maximum concentrations of 100 and 50 µg/mL; however, within 72 h, the tested GM and RAW cells recovered and began to proliferate again. In HeLa cells, in the first 24 h, L-asparaginase did not show significant cytotoxic activity. However, after 48 h, L-asparaginase cytotoxicity was observed at concentrations up to 12.5 µg/mL, and within 72 h, cytotoxicity remained at concentrations of 100 and 50 µg/mL.

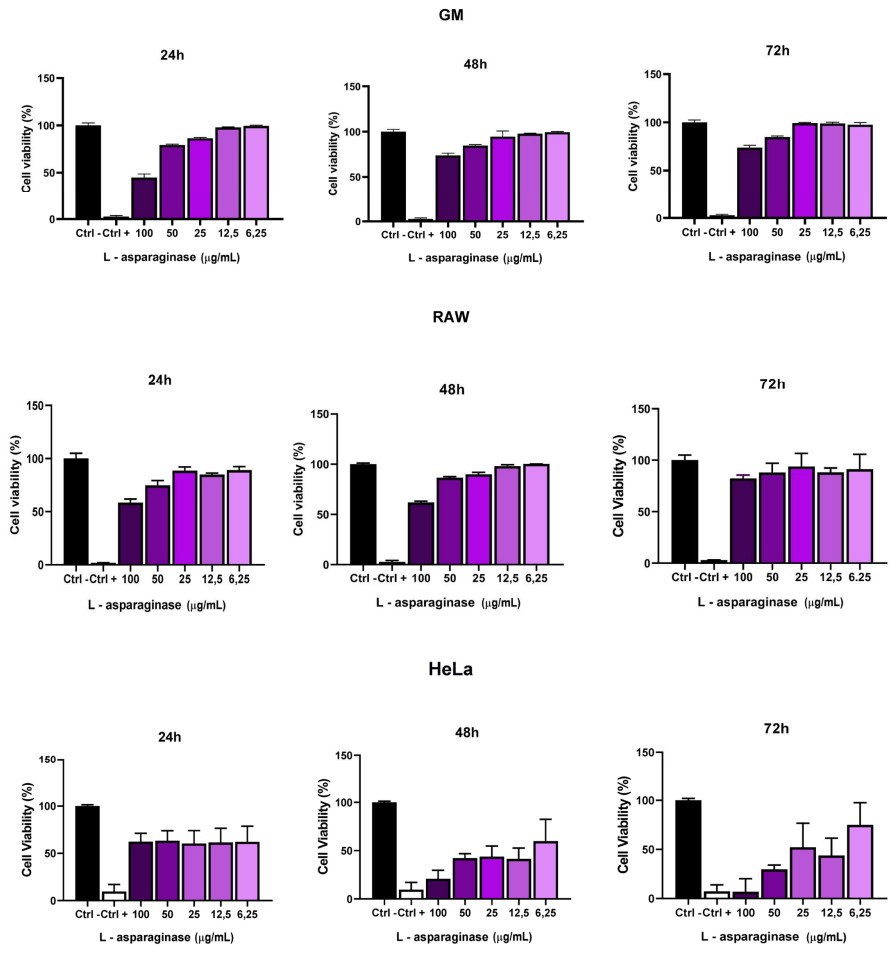

**Figure 5.** Cell viability after treatment with *A. niger* L-asparaginase. RAW cells, GM cells, and HeLa cells. MTT test analyzing the viability of RAW, GM, and HeLa cells after treatment with *A. niger* L-asparaginase. Significant differences ($p < 0.05$) found by one-way analysis of variance (ANOVA). Triplicate tests. Ctrl—(negative control) indicates without treatment with L-asparaginase, and Ctrl+ (positive control) indicates cell death caused by 10% DMSO.

*3.6. Morphological Viability of the HeLa Cell Line under L-Asparaginase Treatment*

The morphology of the HeLa cell line treated with the enzyme L-asparaginase at a concentration of 100 µg/mL after 72 h of treatment was analyzed using an inverted microscope and analyzed using ImageJ software Version 1.8.0 (NIH, Bethesda, MD, USA) to determine the surface area and perimeter. Control HeLa cells (Figure 6A) presented typical morphology with an elongated structure, with growth forming a monolayer and adhesion to each other.

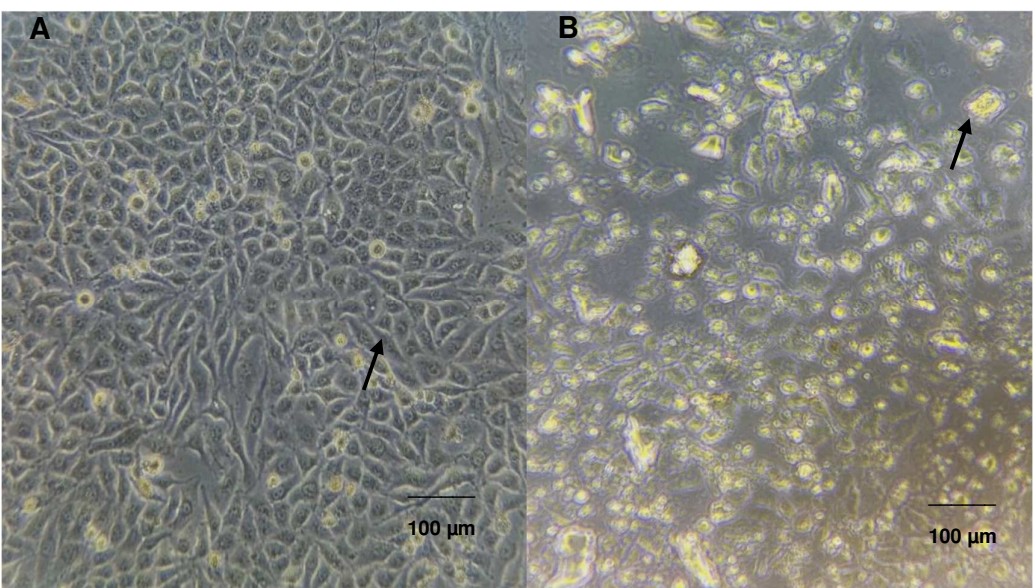

**Figure 6.** Control HeLa cell line without treatment with L-asparaginase (**A**) and after treatment with L-asparaginase from A. niger for 72 h with L-asparaginase with cell death as indicated by the arrow (**B**). HeLa cell line treated with 100 µg/mL in low glucose DMEM medium with 10% fetal bovine serum and 1% antibiotic.

The cells treated with the enzyme (Figure 6B) showed morphological changes compared to the control cells, with circular formations and a reduction in size, an increase in the area without the presence of cells, and a rough appearance on their surface, which may indicate the loss of its cytoplasmic material, indicating that the enzyme affects its morphological structure via changing the exposure time and the characteristics of the enzyme with the cells.

## 4. Discussion

With an enzyme activity of 46.4 U/mL using the semi-solid fermentation method, *A. niger* in this study exhibited higher enzyme activity compared to other fungal genera. Profitability is a crucial aspect to consider when contemplating large-scale enzyme production. Using industrial waste as substrates for enzyme production not only leads to profitable production but also promotes cleaner production methods. To produce L-asparaginase, agro-industrial residues have been reported as suitable substrates in several studies [23,28,29], such as wheat bran in this study. Simulating the natural habitat of *A. niger* facilitates the production of enzymes and minimizes the risk of contamination. It is easy to maintain and allows a high concentration of proteins. This demonstrates that *A. niger*, through semisolid fermentation, can be a new alternative to produce high-activity and high-concentration L-asparaginase [30].

The *A. niger* in this study previously demonstrated potential in the production of L-asparaginase compared to other genera chosen in this work. Studies carried out by Vala et al. [23], Babu et al. [31] and Karanam & Medicherla [32] showed that *A. niger* has a high capacity for L-asparaginase production, significant enzymatic activity, and resistance to various pH levels and temperatures. They obtained results similar to those of this

work, justifying its selection for this study and showing its ability to be an alternative for producing this enzyme on a large scale.

In enzyme purification, the precipitation technique is used to remove the protein of interest from the sample and reduce interactions with other biomolecules that could affect enzyme activity. Therefore, the enzyme can be used in biotechnological processes without interference. Sample preparation becomes a challenge in developing faster methods that require smaller volumes and more efficient protein concentration. Among the commonly used steps, protein precipitation using solvents such as ethanol is often employed due to its high ability to concentrate the target molecule and its profitability, making it part of almost all purification processes [33].

The advantage of ethanol precipitation lies in the fact that the solvent can be rapidly evaporated from the sediment at room temperature, eliminating the need for subsequent processes to remove the precipitant [34]. Thus, ethanol proves to be the most suitable precipitant for the precipitation step of L-asparaginase produced by *A. niger*, as demonstrated in the results of this study. Vala et al. [23] also achieved superior results using ethanol to precipitate L-asparaginase for purification purposes. According to El-Naggar [35], any protein, once produced by a biological entity, must be purified to characterize its physical and biological properties. A protein must be free of contaminants before being used in any application. Implementing an enzyme in pharmacological studies or other industrial processes depends on understanding its biochemical conditions and how the surrounding environment affects the enzyme.

The process of biochemical characterization of the enzyme, including optimum temperature, optimum pH, kinetics, and the influence of compounds on its enzymatic activity, is crucial. This characterization ensures that the enzyme can be properly applied in pharmacological compounds without significant loss of activity [36].

The pH of the solution affects the generation of hydroxyl radicals and influences the surface charge and potential interface properties of the catalyst, making it one of the essential factors. Each enzyme has an optimum pH at which its activity is at its maximum [37]. The results obtained in this study were similar to the findings of Vala et al. [23], where purified L-asparaginase from *A. niger* AKV-MKBU obtained from seawater samples had an optimum pH of 9.0. Similar optimum pH values of 8.0 and 9.0 were found for *E. coli* [38].

Regarding temperature and stability, unlike other L-asparaginases produced by *A. niger* AKV-MKBU [23], *Brevibacillus borstelensis* ML12 [39] showed an optimum temperature of 30 °C. *Bacillus* sp. was reported to have an optimum temperature of 40 °C [40] and 37 °C for *Rhizobium* etli [41] but did not exhibit resistance to other temperature ranges. L-asparaginase produced by *Brevibacillus borstelensis* ML12 [39] demonstrated stability at a temperature similar to that of *A. niger* studied in this work. These results highlight the distinct characteristic of L-asparaginase produced by *A. niger*, where it showed greater resistance to different temperatures compared to L-asparaginases produced by other microorganisms. This demonstrates that L-asparaginase produced by *A. niger* has the potential to serve as a new alternative source of this enzyme.

In the presence of surfactants, SDS exhibits a retarding effect on enzyme activity progressively as the concentration is increased to 20 mM, as supported by Krishnapura and Belur [42]. Studies such as L-asparaginase produced by *B. borstelensis* ML12 reported a significant increase in enzymatic activity with Tween 20, probably due to greater exposure of the enzyme's active site due to the reduction in surface tension. Another study by Meghavarnam and Janakiraman reported an increase in L-asparaginase activity from *Fusarium culmorum* [43].

The cytotoxic activity of *A. niger* L-asparaginase was tested in RAW, GM, and HeLa cells in this work. L-asparaginase produced by microorganisms and plants demonstrated cytotoxic potential in vitro against several types of cells, as shown in the study by Asthana et al. [44]. In the research by Rani et al. [45], L-asparaginase produced by *A. flavus* at a concentration of 131.25 µg/mL inhibited about 50% of HeLa cell growth, and in another later study, the dose-dependent oncogenic activity of L-asparaginase produced by *Aspergillus oryzae* occurred up to a concentration of 2 µg/mL in HeLa cells [46].

L-asparaginase produced from microbial cells is capable of reaching HeLa cells, according to Fátima et al. [47], who presented in their research that L-asparaginase isolated from *P. aeruginosa* (P31, P32, and P34) at concentrations 86.7, 40.3, and 57.6 µg/mL, respectively, had IC50 in HeLa cells, with increased cytotoxicity at enzyme concentrations of 5, 10, 25, 50, 75, and 100 µg/mL, with maximum viability of 46.08% and minimum viability of 28.33%.

In the current study, RAW, GM, HeLa, and SiHa cells were evaluated with lyophilized purified L-asparaginase at the following concentrations: 100, 50, 25, 12.5, and 6.25 µg/mL. L-asparaginase in GM and RAW cells showed a slight cytotoxic activity in 24 and 48 h only at maximum concentrations of 100 and 50 µg/mL; however, in 72 h, GM and RAW cells recovered. In HeLa cells, in the first 24 h, L-asparaginase did not show significant cytotoxic activity; however, after 48 h, L-asparaginase cytotoxicity was observed at concentrations up to 12.5 µg/mL, and within 72 h, cytotoxicity remained at concentrations of 100 and 50 µg/mL. L-asparaginase from *A. niger* was able to effectively inhibit the growth of human cervical cancer cells in vitro of HeLa origin, which in the future could form a therapeutic agent capable of treating cervical cancer caused by HPV 18.

The enzyme, the object of this study, also demonstrated significant results with regard to cell viability in HeLa cells, which presented, after the use of 100 µg/mL of L-asparaginase for 72 h, important morphological changes compared to control cells. After the experiment, due to the time of exposure to the enzyme, HeLa cells exhibited circular formations with a significant reduction in size and a rough appearance on their surface, indicating a loss of cytoplasmic material and relevant degeneration of cell bodies, making it possible to see empty spaces in the analyzed fields. Similar results were found in the study by Fátima et al. [47], with L-asparaginase isolated from *P. aeruginosa*, demonstrating that the enzyme could be used as an effective therapeutic agent in the treatment of cervical cancer. However, this requires additional in vivo studies.

In the near future, a significant increase in the demand for L-asparaginase is expected due to its increasing application in both clinical and industrial contexts, covering several sectors of the biotechnology industry. The *A. niger* L-asparaginases purified and characterized in this study proved to be highly efficient and low-cost in this research. Therefore, it is important to explore new sources for L-asparaginase production.

## 5. Conclusions

These results demonstrate the ability of *A. niger* to produce the L-asparaginase enzyme, highlighting its low cost, easy obtainment from the environment, and resistance to different pH levels, temperatures, ions, and surfactants. These findings suggest that this L-asparaginase could serve as an alternative for applications in pharmaceutical and biotechnological processes. Using simple enzymatic purification techniques, this L-asparaginase was obtained with satisfactory purity, high protein concentration, activity, and low cytotoxicity in healthy cells. It was observed that L-asparaginase has the ability to significantly inhibit the proliferation of HeLa cells, even in a short period of enzymatic reaction, indicating that L-asparaginase from *A. niger* may represent a new alternative source for future treatment oncology. However, this requires additional in vivo studies.

**Author Contributions:** Conceptualization, M.d.D.S.B.N.; Methodology, S.M.d.S.D., A.K.d.C.S., K.R.A.B., C.B.C., F.J.L.L., M.A.C.N.d.S., M.d.S.A. and M.d.D.S.B.N.; Validation, S.M.d.S.D., M.A.C.N.d.S. and M.d.S.A.; Formal analysis, S.M.d.S.D. and C.B.C.; Investigation, S.M.d.S.D.; Supervision, M.A.C.N.d.S.; Project administration, M.d.D.S.B.N.; Funding acquisition, M.d.D.S.B.N. All authors have read and agreed to the published version of the manuscript.

**Funding:** The authors express their gratitude to the Federal University of Maranhão (UFMA), the Foundation for Research and Technological Development of Maranhão (FAPEMA), and the Coordination for the Improvement of Higher Education Personnel—Brazil (CAPES: Financing code 001: Procad Amazonia number 23038015702/2018—M.D.S.B.N) for their support.

**Institutional Review Board Statement:** Not applicable.

**Informed Consent Statement:** Not applicable.

**Data Availability Statement:** Data are contained within the article.

**Conflicts of Interest:** The authors declare no conflict of interest.

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
