# Peer review of "Production, Characterization Purification, and Antitumor Activity of L-Asparaginase from Aspergillus niger"

_fermentation, doi:10.3390/fermentation10050226_

Round 1
Reviewer 1 Report
Comments and Suggestions for Authors
The titrated work Production, Characterization, Purification and Antitumor Activity of L-asparaginase from Aspergillus niger investigates the influence of the type of fermentation, precipitant, purification, characterization, and in vitro cytotoxicity of L-asparaginase for use in Hela cell treatment. The research addresses a relevant topic of considerable impact for the scientific community. The results are promising and applicable. However, some important points should be considered for improving research and discussion in general. Authors should consider the following points:
1) Analyzing the results of fermentation with different microorganisms, a question arises that extends throughout the work. Even though the results show that Aspergillus niger produced a L-asparaginase with higher enzymatic activity (46.40 U/mL), the enzyme produced by Fusarium solani showed relatively close activity (37.64 U/mL). It would be interesting if the authors worked in parallel with the production of the enzyme L-asparaginase by both microorganisms. This is because, in the study of the stability and effectiveness of the application in the treatment of the HeLa cell, the authors could conclude that although the L-asparaginase produced by Aspergillus niger showed higher activity during production, the enzyme produced by Fusarium solani showed greater stability and efficiency after purification, being more advantageous its production. However, this would only be possible with the study of both enzymes.
2) All stability tests performed should be redone considering times greater than 1 hour. Since the focus of this work is the production of enzymes to be applied in drugs, it is essential to know the half-life of enzymes to know the optimal dosage for a more effective treatment.
3) Regarding the structure of the text, at the beginning of the Introduction, the authors mention that "cervical cancer is caused by a persistent and high-grade infection, which can be caused by HPV..." but do not link it with HeLa cells. In this sense, a line must be added to this connection. In addition, authors could use justified text formatting instead of separating the words at the end of each line.
Comments on the Quality of English LanguageAuthors could review grammar and punctuation.
Author Response
Thank you very much for taking the time to review our work and for the detailed observations you shared with us. Your suggestions are valuable and contribute significantly to improving the quality of our research.
We would like to address the points that were mentioned:
1) Analyzing the results of fermentation with different microorganisms, a question arises that continues throughout the work. Although the results show that Aspergillus niger produced an L-asparaginase with greater enzymatic activity (46.40 U/mL), the enzyme produced by Fusarium solani presented relatively similar activity (37.64 U/mL). It would be interesting if the authors worked in parallel with the production of the L-asparaginase enzyme by both microorganisms. This is because, in studying the stability and effectiveness of the application in the treatment of HeLa cells, the authors were able to conclude that although the L-asparaginase produced by Aspergillus niger presented greater activity during production, the enzyme produced by Fusarium solani presented greater stability and efficiency after purification, with its production being more advantageous. However, this would only be possible with the study of both enzymes.
Response: We understand your suggestion to study L-asparaginase produced by both microorganisms, however, budgetary limitations made it impossible to analyze the efficiency of L-asparaginases produced by other genera of fungi, considering that the reagents used in our research center are the same ones used by several other research groups, although there are not enough of them. For this reason, we chose to focus our efforts on L-asparaginase produced by Aspergillus niger for the subsequent stages of the research, as it was the species that showed the greatest enzymatic activity.
2) All stability tests carried out must be redone considering times longer than 1 hour. As the focus of this work is the production of enzymes for application in medicines, it is essential to know the half-life of the enzymes to know the ideal dosage for a more effective treatment.
Response: Unfortunately, due to the deadline stipulated by the journal for reviewing and adjusting the manuscript and our limited resources, it was not possible to conduct analyzes of all suggested stability tests. To verify the half-life of the enzyme at temperature stability, tests with sequenced times would be necessary, which would require time and adequate equipment for verification. It should be noted that the existing equipment at the Federal University of Maranhão (UFMA) is limited and in common use for all Undergraduate courses and Postgraduate programs, which made carrying out the tests unfeasible. For these reasons, we used the 1 hour time period judiciously and based on information obtained through recent studies that explored the characterization of L-asparaginase in other similar microorganisms. We fully recognize the importance of these additional analyzes in order to understand the half-life of the enzymes, seeking to identify the ideal dosage for a more effective treatment, which would further enrich the work. However, the scarcity of investments in the area of Teaching and Research in Brazil limited the authors to carry out the present work in full compliance with the suggested recommendations, which is why additional tests will be the subject of future study as a way of complementing the present work.
3) Regarding the structure of the text, at the beginning of the Introduction, the authors mention that “cervical cancer is caused by a persistent and high-grade infection, which can be caused by HPV...” but do not link it to HeLa cells. In this regard, a line must be added to this connection. Additionally, authors could use justified text formatting instead of separating words at the end of each line.
Response: We made the necessary correction, adding a line to establish the textual connection in a clearer and more cohesive way. Regarding the formatting of the text, we sent this justification, we believe there was a misconfiguration at the time of formatting. Thank you for your observation and contribution to improving our work.
Thank you for reviewing our manuscript. We have carefully considered all your comments and suggestions, and have made revisions accordingly. The authors appreciate your understanding regarding these financial and time constraints. Once again, we sincerely thank you for your time, effort, and valuable contributions to our work. Their suggestions were extremely useful and enriching, providing valuable insights for future research. Changes proposed in the manuscript have been highlighted in yellow to facilitate correction. Additionally, we would like to acknowledge that English revision has also been conducted.

Reviewer 2 Report
Comments and Suggestions for Authors
This article discusses Production, Characterization, Purification, and Antitumor Ac-2 Activity of L-asparaginase from Aspergillus niger. The writing of the manuscript is concise and easy to understand; It has certain scientific significance. It is suggested be accept after addressing the minor issues.
1. Attention should be paid to the Latin names of bacterial species in the manuscript. When the same genus of bacteria appears for the second time or later, it is recommended to abbreviate the genus name.
such as, Line 30, Lines 224, 226, ect.
2. The format of references should be consistent; Especially for journal names, it is recommended to use italics.
3. Suggest adding vertical coordinates to Figure 2,beautifying Figure 2.
4. Fig.7, ug?
Comments on the Quality of English LanguageNone.
Author Response
Thank you very much for taking the time to review our work and for the detailed observations you shared with us. Your suggestions are valuable and contribute significantly to improving the quality of our research.
We would like to address the points mentioned:
1) Attention should be paid to the Latin names of bacterial species in the manuscript. When the same genus of bacteria appears for the second time or later, it is recommended to abbreviate the genus name. Such as, Line 30, Lines 224, 226, ect.
Response: Corrections were made, where we followed the recommendations to abbreviate the name of the genus as suggested.
2) The format of references must be consistent; especially for periodical names, italics are recommended.
Response: We made the necessary corrections to ensure uniformity, especially with regard to the use of italics for journal names, as recommended.
3) Suggest adding vertical coordinates to Figure 2, beautifying Figure 2.
Response: We made the suggested changes, aiming to beautify and make Figure 2 more understandable, in addition to adding vertical coordinates with the molecular weights of the standard used.
- Fig.7, ug?
Response: We clarify that the “µg” units were used to represent micrograms of enzyme in the cytotoxicity assay. We chose these units because of their common application in recent studies in the literature for this specific type of essay.
Once again, I sincerely thank you for your time, effort, and valuable contributions to our work. Changes proposed in the manuscript have been highlighted in yellow to facilitate correction. Additionally, English revision has been conducted.

Round 2
Reviewer 1 Report
Comments and Suggestions for Authors
Prezados autores,
Obrigado pelos esclarecimentos referentes às questões sugeridas e às correções realizadas. Acredito que a falta de apoio à pesquisa e desenvolvimento é um desafio a ser considerado. É aceitável justificar a prioridade em experimentos apenas com Aspergillus niger , mas os testes de estabilidade e meia-vida da enzima são essenciais para um trabalho que visa produzir uma enzima para aplicação em medicamentos. Novamente, é essencial conhecer a meia-vida das enzimas para saber a dosagem ideal para um tratamento mais eficaz. Portanto, os autores devem realizar todos os testes de estabilidade considerando tempos superiores a 1 hora.
Parece correto.